# Metagenomic analysis for taxonomic and functional potential of Polyaromatic hydrocarbons (PAHs) and Polychlorinated biphenyl (PCB) degrading bacterial communities in steel industrial soil

**Monika Sandhu[1], Atish T. Paul[2], Prabhat N. Jha[1] ***

**1** Department of Biological Sciences, Birla Institute of Technology and Science Pilani, Pilani, Rajasthan, India, **2** Department of Pharmacy, Birla Institute of Technology and Science Pilani, Pilani, Rajasthan, India

* prabhatjha@pilani.bits-pilani.ac.in

## Abstract

Iron and steel industries are the major contributors to persistent organic pollutants (POPs). The microbial community present at such sites has the potential to remediate these contaminants. The present study highlights the metabolic potential of the resident bacterial community of PAHs and PCB contaminated soil nearby Bhilai steel plant, Chhattisgarh (India). The GC-MS/MS analysis of soil samples MGB-2 (sludge) and MGB-3 (dry soil) resulted in identification of different classes of POPs including PAHs {benzo[a]anthracene (nd; 17.69%), fluorene (15.89%, nd), pyrene (nd; 18.7%), benzo(b)fluoranthene (3.03%, nd), benzo(k) fluoranthene (11.29%; nd), perylene (5.23%; nd)} and PCBs (PCB-15, PCB-95, and PCB-136). Whole-genome metagenomic analysis by Oxford Nanopore GridION Technology revealed predominance of domain bacteria (97.4%; 97.5%) followed by eukaryote (1.4%; 1.5%), archaea (1.2%; 0.9%) and virus (0.02%; 0.04%) in MGB-2 and MGB-3 respectively. Proteobacteria (44.3%; 50.0%) to be the prominent phylum followed by Actinobacteria (22.1%; 19.5%) in MBG-2 and MBG-3, respectively. However, Eukaryota microbial communities showed a predominance of phylum Ascomycota (20.5%; 23.6%), Streptophyta (18.5%, 17.0%) and unclassified (derived from Eukaryota) (12.1%; 12.2%) in MGB-2 and MGB-3. The sample MGB-3 was richer in macronutrients (C, N, P), supporting high microbial diversity than MGB-2. The presence of reads for biphenyl degradation, dioxin degradation, PAH degradation pathways can be further correlated with the presence of PCB and PAH as detected in the MGB-2 and MGB-3 samples. Further, taxonomic *vis-à-vis* functional analysis identified *Burkholderia*, *Bradyrhizobium*, *Mycobacterium*, and *Rhodopseudomonas* as the keystone degrader of PAH and PCB. Overall, our results revealed the importance of metagenomic and physicochemical analysis of the contaminated site, which improves the understanding of metabolic potential and adaptation of bacteria growing under POP contaminated environments.

**Data Availability Statement:** All raw data files of the metagenome reads are available from the NCBI Sequence Read Archive database under BioProject

PRJNA765179 (accession number(s)
SRR16004303 and SRR16004304). All relevant
data are within the manuscript and its Supporting
Information files.

**Funding:** Authors are thankful to Department of
Science and Technology, New Delhi, India for
financial assistance under Women Scientist
Scheme B [SR/WOS-B/570/2016] to MS. https://
online-wosa.gov.in/wosb/.

**Competing interests:** The authors have declared
that no competing interests exist.

## Introduction

Persistent organic pollutants (POPs) are anthropogenic chemicals that are enlisted as priority environmental pollutants due to their toxicity and persistence in the environment for a prolonged period [1]. PAHs/PCBs are strongly lipophilic and hence they easily enter the food chains. These characteristics are important since they are responsible for the detrimental effect on the environment and induce health threats to plants, animals, and humans [2]. Increased industrialization has led to the extensive production of such POPs, which are also emitted during steel production [3]. The rise in these pollutants has led to adverse health and the environment effects, resulting in extensive studies on the remediation of contaminated soil. Various physical and chemical technologies, including chemical oxidation, electrokinetic remediation, solvent extraction, photocatalytic degradation, and thermal treatment, are widely applied in remediation [4]. However, most of these treatment methods are unsustainable, disruptive and carry these PCBs/PAHs to the environment. Therefore, the utilization of existing contaminated soil-based bacterial communities can be an alternative strategy for effective and viable degradation of POPs [5] as it has comparatively fewer technical hindrances than other remediation technologies.

A series of studies have been performed by the culture-dependent approach to isolate the most efficient biodegrader from such polluted sites [6]. The contaminated soil environment consists of the genetic, species, and metabolic diversity of microbial biodegraders. Only a minor fraction of POP-degrading bacteria can be obtained using a culture-dependent method o. Furthermore, it has been reported that enrichment of these cultures under lab conditions is less efficient in biodegradation than indigenous bacteria present in the contaminated soil [7]. To date, information related to taxonomic and functional interaction amongst the microbial communities during the biodegradation process within the contaminated environment is skewed. The recent development of powerful culture-independent metagenomic approaches and the advancement of next-generation sequencing (NGS) technology provides a comprehensive insight into the total microbial community inhabiting contaminated sites and their metabolic capabilities. Several metagenomics studies conducted on PAH/PCB contaminated soil samples [8] have highlighted microbial interaction playing a pivotal part in bioremediation of these POPs. However, most of these studies are based on 16S rRNA gene sequence, which does not highlight the metabolic potential of resident microorganisms. Therefore, the present study aimed to investigate taxonomic diversity and their metabolic potential to degrade POPs employing Oxford Nanopore Technology (ONT). ONT is very sensitive and can detect very low abundant microbial members that are otherwise missed in the metagenome.

The PCBs congeners and PAHs have been reported to be present in the waste sites of the industrialized area of this steel plant, India [9]. The major pollution source of steel industries includes sinter, coke, and the blast furnace [10]. Therefore, the present study aimed at investigating and providing an insight into the diversity of the bacterial community, metabolic potential of the dominant bacterial community in the contaminated soil collected from nearby regions of Bhilai steel plant (one of Asia's biggest steel plants) in Chhattisgarh, India, and to correlate their functional characteristics to the biodegradation pathways. From our result, we concluded that our results provided potential bacterial candidates for the exploitation bioremediation of PAH and PCB.

## Materials and methods

### Study site and sampling

The soil samples were collected from 2 different sites i.e., sludge site (Metagenomics Bhilai; MGB-2) and dry soil waste site (Metagenomics Bhilai; MGB-3) from the polluted area near

Bhilai steel plant, Chhattisgarh (21.1915˚ N, 81.4041˚ E), in India. Soil samples were collected in sterile containers from a depth of about 0 to 10 cm of two sampling sites. The soil samples were randomly collected from three sites of each site and pooled for further analysis. It was then transported on ice pack and stored at 4˚C (to be used immediately) in the lab for analysis. Physicochemical parameters such as pH, electrical conductivity, organic C, N, P, Mg, K, Na, Cl, Ca, S, Zn, Fe, Cu, and Mn of MGB-2 and MGB-3 were estimated using the standard protocol at the National Horticultural Research and Development Foundation, Nasik, India.

## Extraction and determination of PAH and PCB in sediments

PCB and PAH were extracted following He et al. [11] with minor modification. Briefly, 5 g of collected sample (dry weight) was added into 50 ml Milli Q (MQ) water and was homogenized by vortexing for 15 min. After allowing it to stand for 30 min, 10 ml of acetone and hexane (1:1; *v/v*) were added to the falcon and vortexed for 3 min. 2g NaCl was added and shaken vigorously for a few min. It was then centrifuged at 4000 x g for 5 min. The supernatant was subjected for further solid-phase extraction (SPE) of PCB and PAH using bond elute cartridge as per manufacturer's instruction (Agilent technologies, USA). Further, the sample elution was performed with methanol and hexane 1:1 (*v/v*) in 5 ml MQ by centrifuging for 2 min at 1000 x g. The final elute was then collected through a Polytetrafluoroethylene (PTFE) filter (0.22 μ) in a separate vial and adjusted to 1 ml with nitrogen [12].

GCMS-TQ8040 (Shimadzu, Japan) fitted with Scan/SIM was used to qualitatively analyze PAHs and PCBs that are potentially present in the MGB-2 and MGB-3 samples. GC-MS/MS fitted with Flame ionization detector (FID), and an RTX-5 column (30 m × 0.32 mm × 0.25 μm) was used for analysis. GC conditions were set at 40˚C with a 2 min hold and 10˚C/min increment to 80˚C, then 6˚C/min to 225˚C with 10 min hold. The presence of PCB was detected through SIM mode of GC-MS/MS.

## Metagenome sequencing and analysis

**DNA extraction and processing for metagenome.**    Two soil samples were collected in triplicate and pooled together for each sample. DNA extraction was done using Powersoil® DNA Isolation Kit (Qiagen, USA) following the manufacturer's instructions. The metagenomic DNA was checked for integrity by agarose gel (1%) using a BioRad Gel documentation system and was quantified by Qubit 3.0 Fluorometer (Invitrogen, USA).

**Preparation of library and whole metagenome sequencing.**    Metagenomic DNA extracted from the collected soil (MGB-2 and MGB-3) were end-repaired using NEBnext ultra II kit (New England Biolabs, USA), cleaned up with 1x AmPure beads (Beckmann Coulter, USA). Native barcode ligation was performed with NEB blunt/TA ligase using NBD103 and cleaned with 1x AmPure beads. Qubit quantified barcode ligated DNA samples were pooled at an equimolar concentration to attain a 1 μg pooled sample. Adapter ligation (BAM), cleaning of library mix and elution of sequencing library was done as per Kumar et al., 2021 [13] and was further used for whole-genome sequencing. The whole-genome library was prepared by using a Native Barcoding kit (EXP-NBD103). Barcode sequences are detailed in the (S1 Table). The sequencing was performed using SpotON flow cell (R9.4) on MinKNOW 2.1 v18.05.5 with a 48 h sequencing protocol [14] on GridION X5 (Oxford Nanopore Technology (ONT, UK).

## Data processing and analysis

The Nanopore raw reads (*'fast5'* format) were base-called (*'fastq5'* format) and demultiplexed using Albacore v2.3.1 and were uploaded to MG-RAST server (version 4.0.3) for taxonomic and functional analysis. Functional annotation by SEED subsystems helps in predicting the

abundance of genes assigned to metabolic pathways in soil. The sequenced reads were interpreted using a multisource non-redundant ribosomal RNA database for taxonomic diversity. They were determined using the contigLCA algorithm against the M5NR database for samples analyzed *via* whole-genome sequencing (WGS) (MG-RAST metagenome MGB-2 and MGB-3 identification numbers = mgm4822000.3, mgm4822001.3). The interpretation was based on E-value cut-off = 1 x $e^{-5}$ and sequence identity of 60% [15, 16]. Raw reads of whole genome metagenome shotgun sequence of two sample MGB-2 and MGB-3 were deposited to the NCBI Sequence Read Archive under BioProject PRJNA765179 with the accession numbers SRR16004303 and SRR16004304, respectively.

### Statistical analysis

Various alpha diversity indices were calculated to study species richness and evenness of the MGB-2 and MGB-3 using PAST4.03 software. The principal component analysis (PCA) plot was constructed using Bray-Curtis matrices with R studio v3.1.2. Comparison of samples MGB-2 and MGB-3 was done using Statistical Analysis of Metagenomic Profiles software [17] with a two-sided G-test (w/Yates'+ Fischer's). Comparative metagenome analysis was done mainly with RefSeq and SEED subsystem to obtain genus/functional abundance, respectively. Cytoscape software v3.7.1 was used to generate networking plots for the study of the interaction of microbial communities of MGB-2 and MGB-3 involved in xenobiotic biodegradation pathways.

## Results and discussion

### Physico-chemical analysis of the MGB-2 and MGB-3

Microbial community structure and function are determined by various environmental factors, including nutritional status and other parameters, such as pH, salinity, presence of metals, and various physicochemical parameters. Therefore, the physicochemical properties of the MGB-2 and MGB-3 were determined and are summarized in Table 1. The sample MGB-3 was richer in terms of macronutrients such as carbon (C), nitrogen (N), and phosphorus (P) which greatly influenced the composition of the microbial community. The organic carbon content, representing the energy flow in the carbon cycle, was 0.85% (slightly high) in MGB-2 and 1.39% (high) in MGB-3 compared to reference values. Our results indicated high carbon content in the given samples because of aromatic organic hydrocarbons present in the contaminated soil. Industrial soil and effluent are considered sources of organic contaminants including POPs like PAHs [18] and PCBs. Because of the hydrophobic nature of these POPs, they tend to bind with the soil and hence add to the organic carbon content of the soil. The sample MGB-3 had very high nitrogen and phosphorus content (N, 734 kg ha$^{-1}$; P, 56.9 kg ha$^{-1}$ respectively) whereas MGB-2 had moderate nitrogen (430 kg ha$^{-1}$) and low phosphorus (17.66 kg ha$^{-1}$) content. MGB-3 exhibited high P and low C/P ratios, indicating the possibility of higher microbial diversity than the MGB-2. Several studies have confirmed that the soil with high microbial diversity has high P and low C/P ratios, while the environment with less P and high C/P ratios shows a low microbial diversity [19]. A higher level of these micronutrients in MGB-2 can be due to the high water content in the sample. Further, these results indicate that both MGB-2 and MGB-3 can support diverse microbial communities.

Overall, micronutrients, including K, Mg, Na, and Mn, were higher in MGB-2 than MGB-3. The level of K (1232 kg ha$^{-1}$) and Mg (768 kg ha$^{-1}$) was found to be higher than the reference value in MGB-2. The availability of inorganic nutrients serves for structural as well as catalytic functions. Therefore, the bacterial communities' total taxonomic and functional profile is predominantly driven by the availability of C and N and the presence of inorganic nutrients, i.e., Ca, K and Mg, to some extent [20]. The result of the metal analysis indicated that the MGB-3

**Table 1. Physio chemical parameters of the contaminated soil sample MGB-2 and MGB-3 from polluted near Bhilai steel plant.**

| Soil testing parameter | MGB-2 | MGB-3 | Reference value |
|---|---|---|---|
| **pH** | 7.83 | 7.69 | V. acidic < 5.0, Acidic 5.0 < 6.0, Normal 6.0–8.0, Alkaline 8.0 < 9.0 |
| **Electrical Conductivity (dSm$^{-1}$)** | 0.21 | 0.183 | <1.0 Normal |
| **Organic Carbon (%)** | 0.85 | 1.39 | < 0.5 Low, 0.50–0.75 Medium, > 0.75—High |
| **Nitrogen (kg ha$^{-1}$)** | 430 | 734 | < 280 Low, 281–560 Medium, > 560 High |
| **Phosphorous (kg ha$^{-1}$)** | 17.66 | 56.9 | < 22 Low, 23–56 Medium, > 56 High |
| **Potassium (kg ha$^{-1}$)** | 1232 | 336 | < 112 Low, 113–280 Medium, > 280 High |
| **Calcium Carbonate (%)** | 7.2 | 7.6 | < 1 Low, 1–5 Normal, 5–10 Sufficient, > 10 Harmful |
| **Available Calcium (ppm)** | 640 | 960 | < 500 Low, 500–1000 Normal, >1000 Sufficient |
| **Magnesium (ppm)** | 768 | 576 | < 250 Low, 250–500 Normal, >500 Sufficient |
| **Available Sodium (ppm)** | 138 | 126.5 | Up to 400 Normal, 400–700 Problem may occur, > 700 Harmful |
| **Chloride (ppm)** | 11.92 | 7.95 | Up to 350 Normal, 350–1050 Slightly problem, > 1050 Harmful |
| **Sulphur (mg kg$^{-1}$)** | 16.37 | 25.37 | < 10 Low, 10–50 Normal, 50 High |
| **Zinc (mg kg$^{-1}$)** | 1.218 | 3.267 | < 0.6 Low, 0.61–5.0Medium, >5.1 High |
| **Iron (mg kg$^{-1}$)** | 8.372 | 14.98 | < 4.5 Low, 4.6–24 Medium, >25 High |
| **Copper (mg kg$^{-1}$)** | 1.286 | 1.575 | < 0.2 Low, 0.3–1.5 Medium, >1.5 High |
| **Manganese (mg kg$^{-1}$)** | 44.12 | 17.44 | < 2.0 Low, 2.1–29 Medium, >30 High |
| **Water Holding Capacity (%)** | 46 | 43.81 | < 20 Low, 20–50 Medium, > 50 High |

had a comparatively higher metal content (Zn, Cu, and Fe) than the MGB-2 (Table 1). The result also showed a high percentage of Mn (44.12 mg kg$^{-1}$) in MGB-2 and Fe (14.98 mg kg$^{-1}$) in MGB-3, providing a suitable environment for microbes to undergo anaerobic biodegradation. The high metal content in both samples could be due to the additives used in a steel factory. The presence of the metals in a soil sample can limit many microbial species, whereas they can support the survival and growth of metal tolerant species. The presence of metals is in accordance with several reports highlighting heavy metals and organic contaminants such as PAH/ PCB from iron and steel industrial soil sites [21]. In addition, Mn (IV) and Fe (III) are known to act as terminal electron acceptors that efficiently remove aromatic compounds from the soil. Fe is the most widely found cofactor involved in deoxygenation reactions in biodegradation studies. It has been reported that Fe containing dioxygenases is incorporated into the active site either as iron centre, Rieske [2Fe-2S] cluster or as heme prosthetic group during PAH and PCB biodegradation [22].

## Determination of PAH and PCB residues in MGB-2 and MGB-3

Cities with long industrial history contribute to the addition of organic pollutants in soil [23, 24]. Hence it was deemed fit to estimate the level of POPs mainly PAHs and PCBs, in given soil samples by GC-MS/MS using Scan/SIM mode. GC-MS/MS triple quadrupole allows detection at very low (femtogram) limits in the matrix through the use of even greater selectivity with selected reaction monitoring (SIM) mode. Based on the data obtained from GC-MS/MS analysis, various PAHs and PCB in industrial soil samples were identified. The structure and the relative percentage abundance of PAH identified in MGB-2 and MGB-3 are given in Fig 1A. F (15.89%) and BkF (11.29%) were found to be dominant PAH species in MGB-2, while Pyr (18.7%) and BaA (17.69%) in MGB-3. High-molecular-weight (HMW; 4–6 rings)

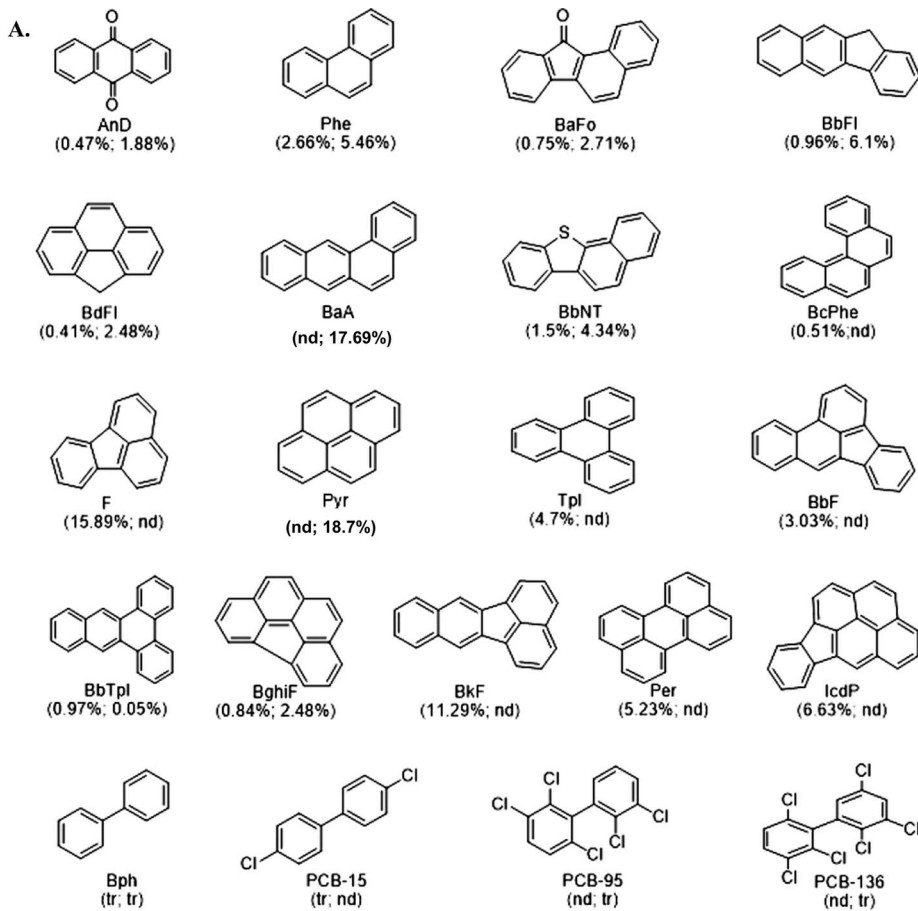

*nd indicates not detected in the given sample
#tr indicates detected in traces

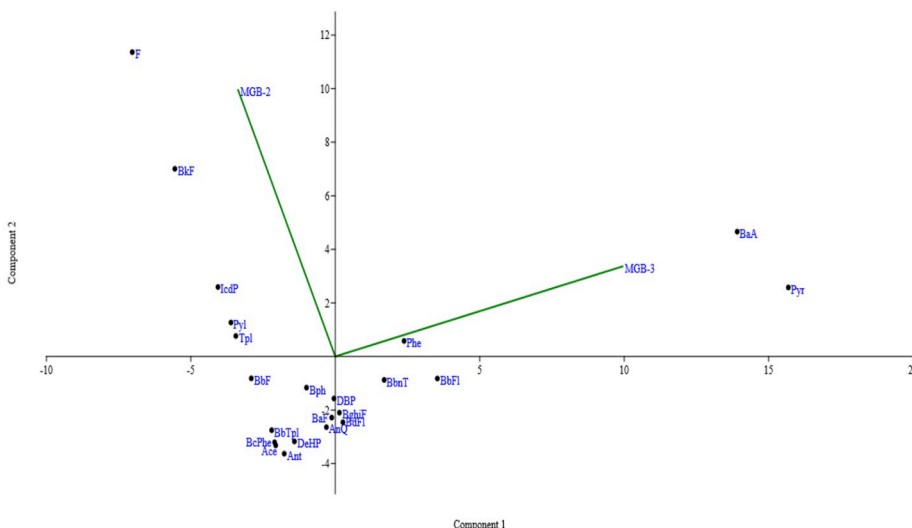

**Fig 1. Structure and relative abundance of PAH and PCB identified in MGB-2 and MGB-3.** (A) PAH found in MGB-2 & MGB-3 were 9,10-Anthracenedione (AnD), 11H-Benzo[a]fluoren-11-one (BaFo), 11H-Benzo[b]fluorene (BbFl), Benzo[def]fluorene (BbFl), Benzo[b]fluoranthene (BbF), Benzo[b]naphtho[2,1-d]thiophene (BbNT), Benzo[b]

triphenylene (BbTPl), Benzo[c]phenanthrene (BcPhe), Benzo[ghi]fluoranthene (BghiF), Benzo[k]fluoranthene (BkF), Fluorene (F), Indeno[1,2,3-cd] pyrene (IcdP), Perylene (Per), Phenanthrene (Phe), Pyrene (Pyr) and Triphenylene (Tpl) and are classified based on the numbers of aromatic rings 2, 3, 4, 5 and 6 membered rings. Bph, PCB-15, PCB-95 and PCB-136 represents biphenyl, 1,1'-biphenyl, 4,4'-dichloro, biphenyl, 1,1'-biphenyl 2,2',3,',5,6 pentachloro and 1,1'-biphenyl 2,2',3,3',6,6' hexachloro respectively. (B) PCoA plot depicting relative abundance using Bray-Curtis matrix of PAH in MGB-2 and MGB-3.

PAHs namely BcPhe (0.51%), BkF (11.29%), F (15.89%), IcdP (6.63%), Per (5.23%), and Tpl (4.70%), predominated in the MGB-2 sediments, which were not detected in MGB-3 (Fig 1B). Similar to our results, PAHs such as Indeno (1, 2, 3-cd) pyrene, benzo (k) fluoranthene, dibenzo (a, h) anthracene, chrysene, fluoranthene, acenaphthene, and fluorene have also been reported to be abundantly present in different operational units of steel industry [25].

In addition to PAHs, MGB-2 and MGB-3 were also found to be contaminated with PCBs such as biphenyl, 1,1'-biphenyl, 4,4'-dichloro (PCB-15) in MGB-2 and biphenyl, 1,1'-biphenyl 2,2',3,5',6 pentachloro (PCB-95) and 1,1'-biphenyl 2,2',3,3',6,6' hexachloro (PCB-136) in MGB-3 (S1 and S2 Figs) [26]. It is well studied that the carcinogenic risk increases with the molecular weight or the aromatic ring of PAHs [27] and with the chlorine atoms in the case of PCBs. However, the data of source of PCBs has been scarce in industrial regions. The present results corroborate with previous studies conducted on the Bhilai steel plant (Raipur, Chhattisgarh) soil, reporting the presence of PCBs ranging from di-chlorinated to hexachlorinated biphenyl in the sludge [28]. From the data of organic contaminants, it appears that the high carbon content in MGB-2 and MGB-3 mentioned in the previous section could be co-related with the abundance of PAH and PCB in the soil sample.

## Metagenomic analysis

**Whole genome sequencing and assembly summary.** The metagenomic approach provides a complete picture of biodegradation *vis-a-vis* microbes present within the environment and the functional genes involved in the bioremediation of contaminants. Whole-genome metagenomics studies are used not only to study the taxonomic diversity but also to elucidate the metabolic pathways required for understanding pollutant degradation [29]. In the present study, the ONT platform for community analysis was used to enable unbiased assembly of complete genome sequencing [30]. Nanopore GridION X5 generated real-time, long-read, high-fidelity DNA sequence data. MG-RAST statistical analysis of dataset provided 275,844 sequences (totalling 539,360,072 bp; average length 1,955 bp) for MGB-2 and 193,221 (totalling 532,031,111 bp; average length 2,753 bp) for MGB-3. The downstream analyses of the total number of reads are detailed in S2 Table. The datasets were used for various taxonomic, ecological, and functional analyses as described in the previous section.

**Analysis of sequence data for the extent of microbial diversity.** The horizontal rarefaction curve indicated the significant sampling depth, representing sufficient sample coverage for diversity analysis (S3 Fig). MGB2 and MGB-3 comprised 1839 and 1884 species, respectively, indicating higher species richness in MGB-3 that is also evident from the data of Chao-1 and Shannon's diversity. The various diversity indices depicted in Table 2 suggest equivalent overall diversity that considers both richness and abundance. Beta diversity among MGB-2 and MGB-3 based on Bray-Curtis dissimilarity (Fig 2A) defined the overall distribution pattern of bacterial communities in MGB-2 and MGB-3 samples, with Principal coordinates showing 98.7%, bacterial communities' similarity in the two sites (MGB-2 and MGB-3).

**Taxonomic diversity and total microbial community.** The collected samples were explored for taxonomic diversity and functional potential. ONT and MG-RAST database search provided detailed microbial taxonomy of MGB-2 and MGB-3. The differences were

**Table 2. Alpha diversity for bacterial, archaea and eukaryota communities of collected soil from polluted site near Bhilai steel plant.**

| Domain | Bacteria | | Archaea | | Eukaryota | | Total Diversity | |
|---|---|---|---|---|---|---|---|---|
| Sample | MGB-2 | MGB-3 | MGB-2 | MGB-3 | MGB-2 | MGB-3 | MGB-2 | MGB-3 |
| **Simpson_1-D** | 0.9957 | 0.9964 | 0.9788 | 0.9798 | 0.9847 | 0.987 | 0.9959 | 0.9966 |
| **Shannon_H** | 6.175 | 6.25 | 4.12 | 4.139 | 4.638 | 4.743 | 6.267 | 6.341 |
| **Evenness_e^H/S** | 0.3456 | 0.3689 | 0.7075 | 0.7298 | 0.4921 | 0.5338 | 0.3057 | 0.3232 |
| **Brillouin** | 6.155 | 6.23 | 4.03 | 4.034 | 4.492 | 4.614 | 6.244 | 6.318 |
| **Menhinick** | 3.247 | 3.168 | 1.849 | 2.024 | 4.17 | 3.856 | 3.972 | 3.91 |
| **Margalef** | 114.7 | 115.1 | 11.17 | 11.34 | 26.66 | 26.61 | 141.9 | 143.6 |
| **Equitability_J** | 0.8532 | 0.8624 | 0.9225 | 0.9293 | 0.8674 | 0.8831 | 0.841 | 0.8488 |
| **Fisher_alpha** | 204.5 | 204.4 | 18.06 | 18.8 | 54.35 | 52.45 | 262 | 264.4 |
| **Berger-Parker** | 0.02676 | 0.0224 | 0.05917 | 0.04873 | 0.04574 | 0.04471 | 0.02608 | 0.02184 |
| **Chao-1** | 1414 | 1429 | 87 | 86.33 | 266 | 267.1 | 1839 | 1884 |

noticeable at the domain level, where MG-RAST derived reads accounted majorly for bacteria (97.4%; 97.5%) followed by eukaryote (1.4%; 1.5%), archaea (1.2%; 0.9%) and virus (0.02%; 0.04%) in MGB-2 and MGB-3 respectively. The abundance of bacteria at the level of phylum, order, class, family, and genus in two samples is shown in Krona plot (Fig 2B and 2C). Among bacteria, phylum Proteobacteria (45.0%; 50.0%) was the most predominant followed by Actinobacteria (22.1%; 19.5%) and Firmicutes (6.3%; 5.4%) in both MGB-2 and MGB-3 samples. Both samples included all six classes of Proteobacteria namely Alphaproteobacteria (18.6%; 22.4%), Betaproteobacteria (8.8%; 10.7%), Deltaproteobacteria (8.2%; 7.1%), Epsilonproteobacteria (0.3%; 0.3%), Gammaproteobacteria (7.2%; 8.9%) and Zetaproteobacteria (0.04%; 0.03%) with difference in their relative abundance, which indicates the presence of diverse members playing a vital role in photosynthesis, nitrogen fixation, sulfur, and phosphorus metabolism [31]. At the class level, Actinobacteria (26.2%) was observed to be dominant in MGB-2 while Alphaproteobacteria (22.1%) in MGB-3. STAMP analysis with G-test (w/ Yates'+ Fischer's) two-way comparison showed the degree of closeness and difference across MGB-2 and MGB-3 at the class level. The analysis showed close similarity in both metagenomes; however, they differ in relative abundance. Among the order, Actinomycetales (85.7%; 88.4%) were distinctly observed in MGB-2 and MGB-3, followed by Solirubrobacterales (8.6%; 6.7%).

At the genus level, the composition of the bacterial community in the two samples was similar, but they differed in terms of their relative abundance. *Streptomyces* (3.7%) and *Candidatus* Solibacter (2.7%) were the most predominant genera in MGB-2, while in MGB-3, it was *Mycobacterium* (3.1%) and *Streptomyces* (3.0%). RefSeq annotations at highest taxonomic classification, i.e., at genus level possess significant abundance level with $p < 1\,e^{-5}$ for *Streptomyces*, *Candidatus* Koribacter, *Gemmata*, *Conexibacter*, *Anaeromyxobacter* and *Nocardioides* in MGB-2, while *Hypomicrobium*, *Pseudomonas*, *Mycobacterium*, *Roseomonas*, and *Methylobacterium* with $p < 1\,e^{-5}$ in MGB-3. The correlation coefficient ($r^2$) indicated a linear relationship between MGB-2 and MGB-3, as depicted on the scatter plot (S4 Fig). The dominance of *Mycobacterium* in MGB-3 can be correlated with the presence of pyrene as the given genus is known to degrade pyrene [32]. The presence of the dominant genera in samples suggests the potential for degrading organic pollutants by the members of these genera. For instance, *Anaeromyxobacter* is known to be capable of anaerobic respiration and possesses genes involved in reductive dechlorination processes at contaminated sites [33]. Similarly, members of *Mycobacterium*, *Bradyrhizobium*, *Burkholderia* [34, 35], and *Pseudomonas* [36, 37] are well-known aromatic pollutant degraders in contaminated soils [38].

**A.**

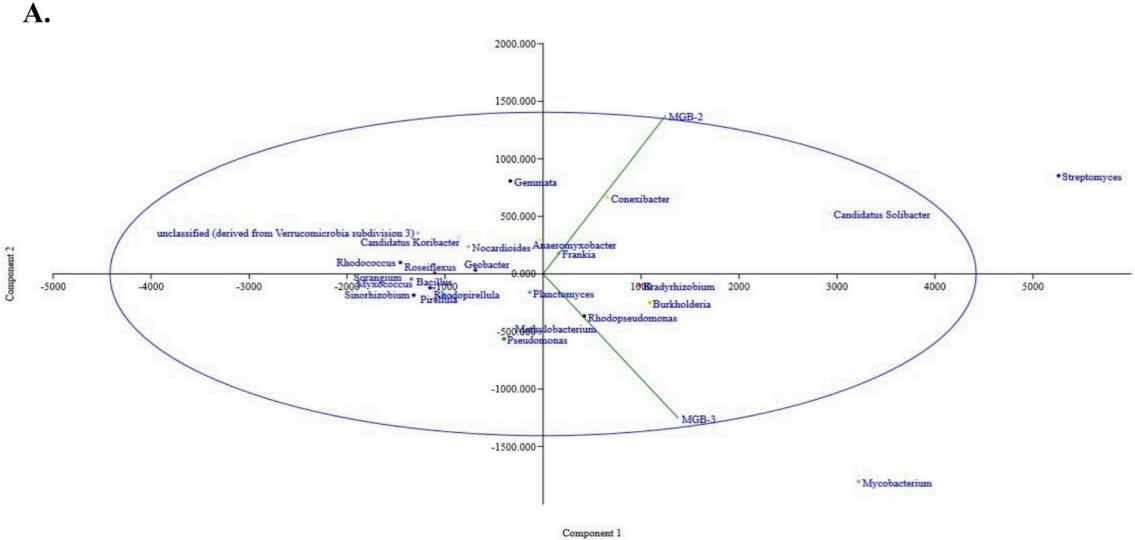

**B.**                                                      **C.**

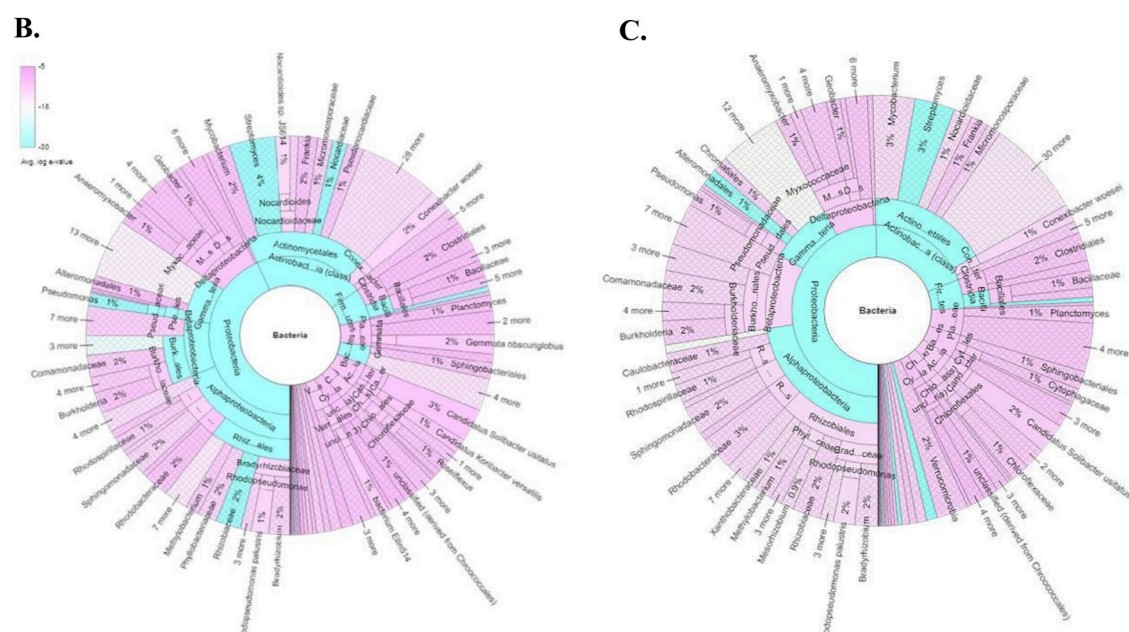

**Fig 2. (A).** Taxonomic analysis beta diversity of MGB-2 and MGB-3 on Bary -Curtis dissimilarity. **(B)** Krona plot demonstrated the relative abundance of bacterial taxa across phylum to genus level hierarchy of a. MGB-2; b. MGB-3.

Within the metagenome of MGB-2 and MGB-3, Archaeal communities showed a dominance of Euryarchaeota (76.7%; 81.3%) at phylum level followed by Crenarchaeota (13.7%; 13.2%) Thaumarchaeota (8.7%; 4.8%) and Korarchaeota (0.9%; 0.7%). At genus level relative abundance for methane-producing *Methanosarcina* (10.5%; 12.1%) *Methanoculleus* (3.7%; 3.2%) ammonia-oxidizing *Nitrosopumilus* (5.9%; 3.3%), Sulfur reducing bacteria like *Thermococcus* (4.3%; 4.7%) unclassified (derived from Euryarchaeota) (4.3%; 4.2%), *Pyrococcus* (3.9%; 3.3%), *Sulfolobus* (3.6%; 3.3%) dominated in both MGB-2 and MGB-3. However, a two-way statistical comparison of Archaea at genus level revealed *Nitrosopumilus* (p = 1.24 e$^{-4}$) and

*Cenarchaeum* (p = 5.18 $e^{-3}$) in MGB-2 while *Methanobrevibacterium* (p = 0.012) in MGB-3 to predominate indicating more methanogenesis occurring in MGB-3 [39]. The contaminated soil also possesses many members of *Methanosarcina*, *Halobacterium*, *Euryarchaeota*, and *Crenarchaeota* of uncultured genera. The diversity of Archaea is found to be higher within hydrocarbon-degrading communities in the contaminated environment than the non-contaminated counterpart [40]. Relative abundance of *Archaeal* communities were more in MGB-2, which was associated with the higher soil moisture and POPs content.

Eukaryota microbial communities showed a predominance of phylum Ascomycota (20.5%; 23.6%), Streptophyta (18.5%, 17.0%) and unclassified (derived from Eukaryota) (12.1%; 12.2%) in MGB-2 and MGB-3. Several studies conducted on contaminated soils have revealed the presence of Ascomycota as the most dominant eukaryotic phylum, as the indigenous ascomycete can transform or remove the pollutants [41].

**Functional diversity and metabolic potential of MGB-2 and MGB-3.**   The SEED subsystem analysis in MG-RAST assigned reads based on various functions that identified 60% of the total function constituting metabolisms of amino acid, carbohydrate, energy, lipid, cofactors, vitamins, biosynthesis of glycan, polyketides, terpenoids, xenobiotic biodegradation, and biosynthesis of secondary metabolites. The remaining 40% functions include cellular processes, organismal systems, genetic & environmental processing, and human diseases.

**General metabolism of microbial community.**   Predicted genes from the MGB-2 and MGB-3 metagenome were used as input in the KEGG mapper for functional annotation. Annotated genes involved in methane (Table 3A), N, and S metabolism pathways were identified (Table 3B). Comparative analysis of both the metagenome based on their metabolism at SEED level 1 revealed significant dominance of methane metabolism (p = 5.61 $e^{-4}$), citrate cycle (p = 1.62 $e^{-3}$), carbon fixation (p = 2.61 $e^{-3}$), pyruvate metabolism (5.80 $e^{-3}$), amino benzoate degradation (p = 0.031), selenocompound metabolism (p = 0.038) and benzoate degradation (p = 0.045) in both samples. However, lipoic acid metabolism (p = 7.58 $e^{-4}$), drug metabolism (p = 7.81 $e^{-3}$), folate biosynthesis (p = 0.17), fatty acid metabolism (p = 0.042) and lysine degradation (p = 0.042) were found to be comparatively high in abundance in MGB-3 (Fig 3A). The results indicated the active metabolism is represented by bacteria belonging to phylum Actinobacteria in MGB-2 while Proteobacteria in MGB-3 (Fig 3B).

The comparative analysis of gene sequences revealed the abundance of P transporter in MGB-2, which can correlate with the low P content than MGB-3. Various genes encoding proteins involved in phosphate-recycling mechanisms, such as *phnA*, *phnE*, *phnW* and *phnX*, (phosphonate transporters), *pstA*, *pstB*, *pstC*, and *pstS* (high-affinity phosphate transporters), and *phoR*, *phoA*, *phoP* and *phoD* (two-component systems) were detected across both the metagenome (S5 Fig). The availability number and abundance of reads for phosphate mechanism in low phosphate-phosphorous environments indicated these mechanisms that help them cope within such environments. Moreover, bacteria can survive due to resistant genes toward toxic metals such as copper, lead, and nickel as a part of their defense mechanism, which is recruited further for cleaning the contaminated environments. The high abundance of Mn content in MGB-2 can be correlated with the significantly high reads for Mn transporter in MGB-2 than MGB-3. Also, a comparison between MGB-2 and MGB-3 of functional gene annotation using SEED subsystem for membrane transport revealed a significant level (p > 0.05) of abundance for $Na^+/H^+$ antiporters and Mn transporter MntH, Mn ABC transporter SitD, TadA, Zn ABC transporter ZnuA (S5 Fig). MGB-3 showed higher $Na^+/H^+$ transporters that/ which indicate the exchange of the ions across the membrane to maintain homeostasis.

**Abundance of Xenobiotics degradation and metabolism gene related to biodegradation.**   Overall, 708 (MGB-2) and 760 (MGB-3) annotated genes corresponding to 17 pathways

**Table 3. Annotated gene and enzyme identified in metagenome MGB-2 and MGB-3.**

**a. methane metabolism**

| Metabolism | sublevel 3 | gene | Enzyme identified |
|---|---|---|---|
| Methane | coenzyme M biosynthesis | *com*E | Sulfopyruvate decarboxylase—beta subunit [EC 4.1.1.79] |
| | coenzyme M biosynthesis | *com*A | Phosphosulfolactate synthase [EC 4.4.1.19] |
| | Hydrogenases | *hox*F | NAD-reducing hydrogenase subunit HoxF [EC 1.12.1.2] |
| | Hydrogenases | *hox*Y | NAD-reducing hydrogenase subunit HoxY [EC 1.12.1.2] |
| | CO Dehydrogenase | *cut*L | Carbon monoxide dehydrogenase form I, large chain [EC 1.2.99.2] |
| | H2:CoM-S-S-HTP oxidoreductase | *hdr*A | CoB—CoM heterodisulfide reductase subunit A [EC 1.8.98.1] |
| | CO Dehydrogenase | *cox*S | Carbon monoxide dehydrogenase small chain [EC 1.2.99.2] |
| | Hydrogenases | *hox*h | NAD-reducing hydrogenase subunit HoxH [EC 1.12.1.2] |
| | CBSS-314269.3.peg.1840 | *cox*L | Carbon monoxide dehydrogenase large chain [EC 1.2.99.2] |
| | CBSS-314269.3.peg.1840 | *cox*M | Carbon monoxide dehydrogenase medium chain [EC 1.2.99.2] |
| | Methanogenesis from methylated | *mtt*B | Trimethylamine:corrinoid methyltransferase [EC 2.1.1.250] |
| | Methanogenesis | *hdr*C2 | CoB—CoM heterodisulfide reductase subunit C [EC 1.8.98.1] |
| | Serine-glyoxylate cycle | *mc*H | N(5), N(10)-methenyltetrahydromethanopterin cyclohydrolase [EC 3.5.4.27] |
| | Methanogenesis | *fhc*D | Formylmethanofuran—tetrahydromethanopterin N-formyltransferase [EC 2.3.1.101] |
| | One-carbon by tetrahydropterines | *mtd*C | Methylene tetrahydromethanopterin dehydrogenase [EC 1.5.99.9] |
| | Methanogenesis | *fwd*A | Formylmethanofuran dehydrogenase subunit A [EC 1.2.99.5] |
| | Methanogenesis | *fno* | N(5), N(10)-methylenetetrahydromethanopterin reductase [EC 1.5.99.11] |
| | Methanogenesis | *fwd*B | Formylmethanofuran dehydrogenase subunit B [EC 1.2.99.5] |

**b. Nitrogen and Sulfur metabolism**

| Metabolism | sublevel 3 | gene | Enzyme identified |
|---|---|---|---|
| Nitrogen | Denitrification | *nir*S | Cytochrome cd1 nitrite reductase [EC 1.7.2.1] |
| | Denitrification | *nir*K | Copper-containing nitrite reductase [EC 1.7.2.1] |
| | Denitrification | *nor*B | Nitric-oxide reductase subunit B [EC 1.7.99.7] |
| | Denitrification | *nor*C | Nitric-oxide reductase subunit C [EC 1.7.99.7] |
| | Denitrification | *nos*Z | Nitrous-oxide reductase [EC 1.7.99.6] |
| | Denitrification | *nor*D | Nitric oxide reductase activation protein NorD |
| | Denitrification | *nor*Q | Nitric oxide reductase activation protein NorQ |
| | Denitrification | *qno*r | Nitric-oxide reductase, quinol-dependent [EC 1.7.99.7] |
| | Nitrate and nitrite ammonification | *nit*H | Polyferredoxin NapH (periplasmic nitrate reductase) |
| | Nitrate and nitrite ammonification | *nir*B | Assimilatory nitrate reductase large subunit [EC:1.7.99.4] |
| | Nitrate and nitrite ammonification | *nit*H | Nitrite reductase [NAD(P)H] small subunit [EC 1.7.1.4] |
| | Nitrate and nitrite ammonification | *nos*Z | Ferredoxin—nitrite reductase [EC 1.7.7.1] |
| | Nitrate and nitrite ammonification | *nrf*E | Cytochrome c-type heme lyase subunit nrfE [EC 4.4.1.-] |
| | Nitrate and nitrite ammonification | *nap*B | Nitrate reductase cytochrome c550-type subunit [EC 1.9.6.1] |
| | Nitrate and nitrite ammonification | *nit*H | Nitrite reductase [NAD(P)H] large subunit [EC 1.7.1.4] |
| | Nitrate and nitrite ammonification | *nap*A | Periplasmic nitrate reductase precursor [EC 1.7.99.4] |
| | Nitrogen fixation | *nif*N | Nitrogenase FeMo-cofactor scaffold and assembly protein NifN |
| | Nitrogen fixation | *nif*E | Nitrogenase FeMo-cofactor scaffold and assembly protein NifE |
| | Nitrogen fixation | *nif*K | Nitrogenase (molybdenum-iron) beta chain [EC 1.18.6.1] |
| Sulfur | Inorganic Sulfur Assimilation | *cys*N | Sulfate adenylyltransferase, dissimilatory-type [EC 2.7.7.4] |
| | Inorganic Sulfur Assimilation | *cys*H | Phosphoadenylyl-sulfate reductase [thioredoxin] [EC 1.8.4.8] |
| | Inorganic Sulfur Assimilation | *si*R | Ferredoxin—sulfite reductase [EC 1.8.7.1] |
| | Inorganic Sulfur Assimilation | *paps*R | Adenylyl-sulfate reductase [thioredoxin] [EC 1.8.4.10] |
| | Inorganic Sulfur Assimilation | *cys*C | Adenylylsulfate kinase [EC 2.7.1.25] |
| | Inorganic Sulfur Assimilation | *cys*I | Sulfite reductase hemoprotein beta-component [EC 1.8.1.2] |

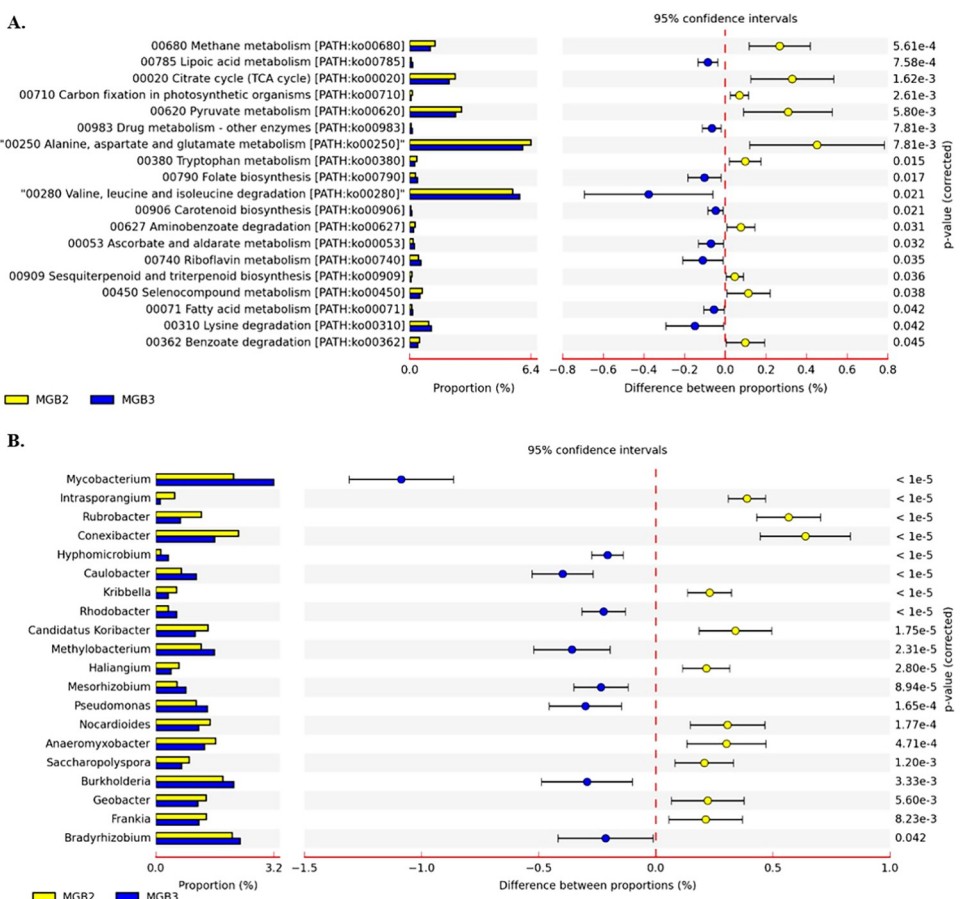

**Fig 3. Comparative analysis of metagenome MGB-2 and MGB-3 using STAMP.** (A) SEED level based on metabolism (B) genus level with RefSeq for metabolism.

linked with xenobiotic biodegradation and metabolism were identified though at varying levels of abundance (S6 Fig) in two samples. The annotated pathways for degradation and metabolism of xenobiotic compounds, including POPs enlisted by the US Environmental Protection Agency, accounted for 2% of the 60% metabolic functions (Fig 4A). The presence of reads for biphenyl degradation, dioxin degradation, PAH degradation pathways can be further correlated with the presence of PCBs and PAHs as detected in the MGB-2 and MGB-3 samples (Fig 4B and 4C). Further, chlorocyclohexane and chlorobenzene [PATH: Ko00361], and benzoate [PATH: Ko00362] degrading pathway were most prominent in MGB-2 (33.76%; 37.39%) and MGB-3 (39.55%; 33.96%). In addition to these degradative pathways, dioxin [PATH: Ko00621] and PAH [PATH: Ko00624] degradation pathways were also observed in MGB-2 (3.27%; 2.54%) and MGB-3 (4.29%; 2.80%), respectively. Similar to our observations, the presence of chlorobenzene, PAH, PCB, and benzoate are known to be prominent contaminants of dye and steel industries [9, 10]. The annotated genes encoding enzymes, namely *chqB*, *pcpB*, *pheA*, *clcD*, *hadL*, *bedC1/todC1*, *bphC*, *catA*, and *catB*, highlighted the degradation of chlorocyclohexane and chlorobenzene compounds within the two communities (Table 4A). The presence of the above-mentioned enzymes suggests that the degradation of chlorobenzene is catalyzed via the ortho -cleavage pathway in two communities [42], and the *meta*-cleavage pathway [43].

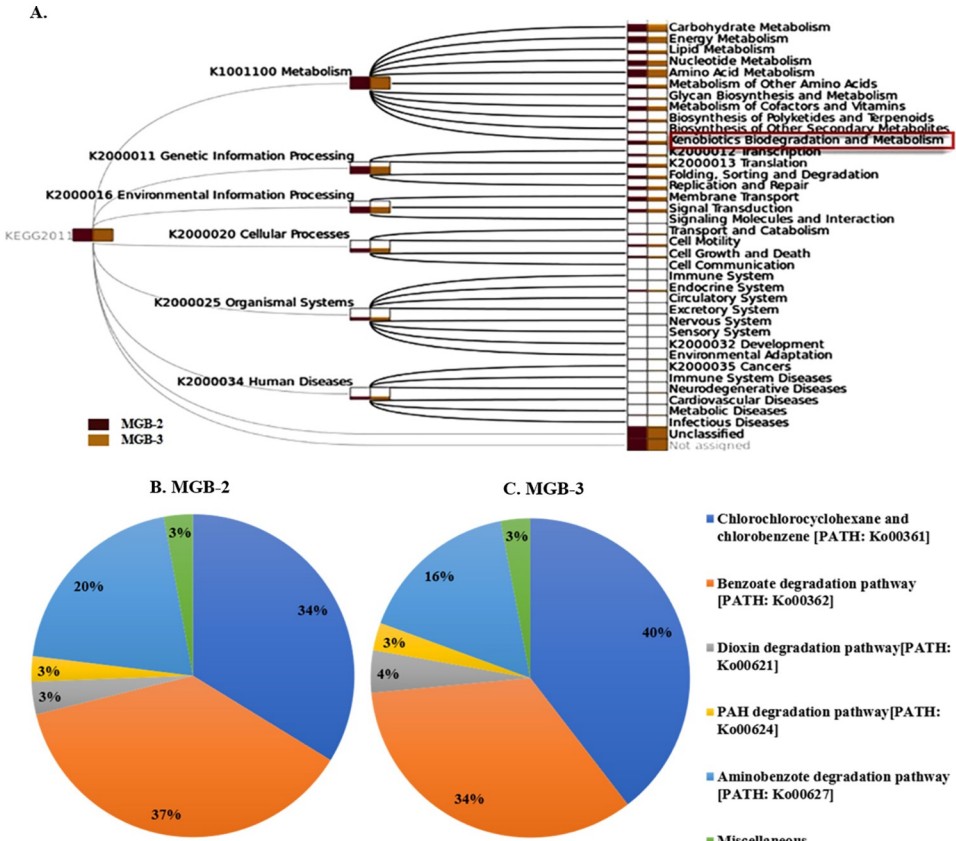

**Fig 4. SEED subsystem analysis in MG-RAST assigned reads in MGB-2 and MGB-3 based on various (A) functions (B) Xenobiotic biodegradation pathways.**

The degradation pathway mainly of PAHs and PCBs is highlighted in the present study as the collected soil samples MGB-2 and MGB-3 were found to be highly contaminated with these organic pollutants. The annotated genes encoding enzymes required in PAH degradation pathway in MGB-2 and MGB-3 are listed in Table 4B. The presence of genes encoding hydroxychromene-2-carboxylate isomerase (*nahD*), naphthalene dioxygenase (*nahAc*), salicylate hydroxylase (*nahG*), and other genes including *nidA*, *nidB*, *nidD*, *phdF*, *phdG*, *phdI*, and *phdJ* indicated the presence of a complete pathway of PAH degradation. The presence of reads for *nidA* gene could be correlated with the degradation of pyrene [5], which was found to be abundantly present in MGB-3 as detected by GC-MS/MS. There are several reports indicating *nid*A gene being responsible for the synthesis of the large subunit of PAH dioxygenase involved in the degradation of PAHs such as phenanthrene, pyrene, benzo[a]pyrene, etc. [47]. The pathway analysis demonstrated that the enzymes involved in the PAH biodegradation pathway were affiliated to the members of genera *Pseudoalteromonas*, *Aromatoleum*, *Dechloromonas*, *Agrobacterium*, *Mesorhizobium* in MGB-2, while *Mycobacterium*, *Parvibaculum*, *Ruegeria*, *Burkholderia*, *Aromatoleum*, *Bradyrhizobium* in MGB-3 communities. The contribution of enzymes for degradation by different genera suggests synergistic degradation of these PAH.

The annotated reads for various genes, including *bphA, bphC, bphD, bphE,* and *bphF*, corresponding to the biphenyl degradation pathway, were identified in both samples in Table 4B. The presence of *bph*A in both metagenomes MGB-2 and MGB-3 indicated biphenyl degradation assigned to genus *Mycobacterium* in both samples and *Polaromonas* in MGB-3. Biphenyl

**Table 4. Annotated enzyme and the assigned genera in metagenome MGB-2 and MGB-3.**

a. Chlorocyclohexane and chlorobenzene degradation pathway

| Pathway | Gene | Anotated Enzyme | Assigned bacteria in MGB-2 | Assigned bacteria in MGB-3 |
|---|---|---|---|---|
| Chlorocyclohexane and chlorobenzene degradation [PATH: ko00361] | chqB | hydroxyquinol 1,2-dioxygenase [EC:1.13.11.37] | Arthrobacter, Bradyrhizobium | Catenulispora |
| | | | Burkholderia, Cupriavidus | Bradyrhizobium |
| | | | Variovorax, Verminephrobacter | |
| | pcpB | pentachlorophenol monooxygenase [EC:1.14.13.50] | Rhodococcus | Anabaena, Burkholderia |
| | pheA | phenol 2-monooxygenase [EC:1.14.13.7] | Arthrobacter, Bradyrhizobium, Nocardia, Renibacterium, Rhodococcus, Rubrobacter | Agrobacterium, Arthrobacter, Azoarcus, Bradyrhizobium, Methylobacterium, Nocardia, Paracoccus, Rhodococcus, Rubrobacter |
| | pnpD | maleylacetate reductase [EC:1.3.1.32] | Bradyrhizobium, Polaromonas | Bradyrhizobium, Burkholderia, Polaromonas, Rhodococcus, Rhodopseudomonas |
| | clcD | Carboxymethylene butenolidase [EC:3.1.1.45] | Acidimicrobium Agrobacterium, Arthrobacter, Azoarcus, Bradyrhizobium, Burkholderia, Candidatus, Cupriavidus, Desulfobacterium, Gemmatimonas, Methanoculleus, Polaromonas, Pseudomonas, Rhodococcus, Rhodopseudomonas, Sphingobium | Acetobacter, Acidimicrobium, Acidobacterium, Agrobacterium, Azoarcus, Beijerinckia, Bradyrhizobium, Burkholderia, Candidatus, Cellvibrio, Cupriavidus, Dechloromonas, Desulfobacterium, Flavobacterium, Gemmatimonas, Polaromonas, Rhodococcus, Sphingomonas |
| | hadL | 2-haloacid dehalogenase [EC:3.8.1.2] | Beijerinckia, Bordetella, Burkholderia, Cupriavidus, Methylobacterium, Mycobacterium, Oligotropha, Rhodopseudomonas | Acidovorax, Anaeromyxobacter, Beijerinckia, Bordetella, Bradyrhizobium, Burkholderia, Candidatus, Chromobacterium, Mesorhizobium, Methylobacterium, Mycobacterium, Methylocella, Polaromonas, Rhodopseudomonas, Sorangium, Xanthomonas |
| | dehH | haloacetate dehalogenase [EC:3.8.1.3] | Anabaena, Anaeromyxobacter, Azoarcus, Burkholderia, Chloroflexus, Cupriavidus, Kribbella, Mesorhizobium, Methylobacterium, Nocardiopsis, Micromonospora Psychromonas, Rhizobium, Rhodopseudomonas, Roseiflexus, Saccharopolyspora | Anabaena, Anaeromyxobacter, Bradyrhizobium, Burkholderia, Chloroflexus, Geodermatophilus, Mesorhizobium, Methylibium, Methylobacterium, Nocardiopsis, Psychromonas, Rhodopseudomonas, Roseiflexus, |
| | bedC1 todC1 | benzene/toluene dioxygenase subunit alpha [EC:1.14.12.3 1.14.12.11] | Mycobacterium | Mycobacterium |
| | bphC | biphenyl-2,3-diol 1,2-dioxygenase [EC:1.13.11.39] | Rhizobium | Mycobacterium, Candidatus, Bradyrhizobium |
| | catA | catechol 1,2-dioxygenase [EC:1.13.11.1] | Cupriavidus, Methylobacterium, Pseudomonas, Sphingobium, Streptomyces | Azospirillum, Bradyrhizobium, Burkholderia, Delftia, Mycobacterium, Pseudomonas, Rhizobium, Sphingomonas, |
| | catB | muconate cycloisomerase [EC:5.5.1.1] | Rhodopirellula | Mycobacterium |
| | dmpB | catechol 2,3-dioxygenase [EC:1.13.11.2] | Arthrobacter, Bradyrhizobium, Brucella, Geobacillus, Meiothermus, Rubrobacter | Agrobacterium, Bradyrhizobium, Burkholderia, Meiothermus, Methylibium, Rhodococcus, |

b. PAH and biphenyl degradation pathway

| Pathway | Gene | Annotated Enzymes | Assigned bacteria in MBG-2 | Assigned bacteria in MBG-3 |
|---|---|---|---|---|
| PAH degradation | nahD | hydroxychromene-2-carboxylate isomerase [EC:5.99.1.4] | Agrobacterium tumefaciens, Aromatoleum aromaticum, Bradyrhizobium sp. BTAi1, Dechloromonas aromatic, Pseudoalteromonas atlantica | Aromatoleum aromaticum, Bradyrhizobium sp. BTAi, Burkholderia xenovorans, Parvibaculum lavamentivorans, Rhizobium leguminosarum, Ruegeria pomeroyi |
| | nahAc | naphthalene dioxygenase ferredoxin [EC:1.14.12.12] | Mesorhizobium loti | nd |
| | nahG | salicylate hydroxylase [EC:1.14.13.172] | Anaeromyxobacter dehalogenans, Burkholderia cenocepacia, Dyadobacter fermentans Methylobacterium extorquens, Mycobacterium smegmatis, Polaromonas sp. JS666, Pseudoalteromonas atlantica | Acidovorax citrulli, Bordetella bronchiseptica, Corynebacterium efficiens, Cupriavidus pinatubonensis, Delftia acidovorans, Polaromonas sp. JS666, Streptomyces coelicolor |
| | nidA | PAH dioxygenase large subunit [EC:1.13.11.-] | nd | Mycobacterium sp. JLS |
| | nidB | PAH dioxygenase small subunit [EC:1.13.11.-] | nd | Mycobacterium sp. JLS |
| | nidD | aldehyde dehydrogenase [EC:1.2.1.-] | Mycobacterium vanbaalenii | Mycobacterium vanbaalenii |
| | phdF | extradiol dioxygenase [EC:1.13.11.-] | nd | Mycobacterium vanbaalenii |
| | phdG | hydratase-aldolase [EC:4.1.2.-] | nd | |
| | phdI | 1-hydroxy-2-naphthoate dioxygenase [EC:1.13.11.38] | Mycobacterium vanbaalenii | Mycobacterium sp. KMS, Mycobacterium vanbaalenii |
| | phdJ | 4-(2-carboxyphenyl)-2-oxobut-3-enoate aldolase [EC:4.1.2.34] | Mycobacterium sp. KMS, Mycobacterium sp. MCS | Mycobacterium sp. MCS |
| Biphenyl degradation | bphA | biphenyl 2,3-dioxygenase [EC:1.14.12.18] | Mycobacterium vanbaalenii PYR-1 | Mycobacterium vanbaalenii PYR-1, Polaromonas naphthalenivorans CJ2 |
| | bphB | cis-2,3-dihydrobiphenyl-2,3-diol dehydrogenase [EC 1.3.1.56] | nd | nd |
| | bphC | biphenyl-2,3-diol 1,2-dioxygenase [EC:1.13.11.39] | Mycobacterium smegmatis str. MC2 155 | Alkalilimnicola ehrlichii MLHE-1, Mycobacterium smegmatis str. MC2 155, Mycobacterium tuberculosis KZN 1435, Mycobacterium bovis BCG str. Tokyo 172 |
| | bphD | 2,6-dioxo-6-phenylhexa-3-enoate hydrolase [EC:3.7.1.8] | Nocardioides sp. JS614 | Mycobacterium marinum M, Polynucleobacter necessaries |
| | bphE | 2-hydroxypenta-2,4-dienoate hydratase [EC 4.2.1.80] | nd | Polynucleobacter necessarius |
| | bphF | 4-hydroxy-2-oxovalerate aldolase [EC 4.1.3.39] | Carboxydothermus hydrogenoformans, Pseudomonas putida F1, Burkholderia sp. 383 | Carboxydothermus hydrogenoformans Z-2901, Legionella pneumophila str. Corby, Mycobacterium sp. JLS, Roseiflexus sp. RS-1, Arcobacter butzleri, Rhodococcus jostii RHA1, Streptosporangium roseum DSM 43021, Nocardioides sp. JS614, Nocardia farcinica IFM 10152, Eubacterium rectale ATCC 33656, Frankia sp. EAN1pec |

(Continued)

**Table 4.** (Continued)

c. Benzoate degradation *via* benzoyl coA, protocatechuate and catechol pathways

| Pathway | Gene | Annotated Enzymes | Assigned bacteria in MBG-2 | Assigned bacteria in MBG-3 |
|---|---|---|---|---|
| Benzoate degradation | *ben*B/ *xyl*Y | benzoate/toulene1,2 dioxygenase beta subunit | *Rhodopseudomonas, Rhodococcus* | *Geodermatophilus, Saccharopolyspora* |
| | *ben*C/ *xyl*Z | benzoate/toulene 1,2 dioxygenase electron transfer component | *nd* | *Geodermatophilus, Mycobacterium, Saccharopolyspora* |
| | *ben*D/ DHBD | 2,3-dihydroxybenzoate decarboxylase [EC:4.1.1.46] | *Burkholderia, Rhodococcus, Rhodopseudomonas* | *Bordetella, Novosphingobium, Polaromonas, Rhizobium, Rhodopseudomonas, Xanthobacter* |
| | *pim*C | pimeloyl-CoA dehydrogenase [EC:1.3.1.62] | *Cupriavidus* | *Cupriavidus* |
| | *pra*C | 4-oxalocrotonate tautomerase [EC:5.3.2.-] | *nd* | *Candidatus, Methylocella* |
| | *ali*A | cyclohexanecarboxylate-CoA ligase [EC:6.2.1.-] | *nd* | *Conexibacter, Cupriavidus, Rhodopseudomonas, Verminephrobacter* |
| | *bad*A | benzoate-CoA ligase [EC:6.2.1.25] | *Achromobacter, Aromatoleum, Azoarcus, Burkholderia, Comamonas, Cupriavidus, Delftia* | *Albidiferax, Amycolatopsis, Aromatoleum, Azoarcus, Bradyrhizobium, Cupriavidus, Leptothrix, Rhodomicrobium* |
| | *bad*D | benzoyl-CoA reductase subunit [EC:1.3.7.8] | *Rhodopseudomonas, Thauera* | *Rhodopseudomonas* |
| | *bad*E | benzoyl-CoA reductase subunit [EC:1.3.7.8] | *Magnetospirillum, Rhodomicrobium* | *nd* |
| | *bad*F | benzoyl-CoA reductase subunit [EC:1.3.7.8] | *Magnetospirillum, Rhodomicrobium, Rhodopseudomonas, Thauera* | *Magnetospirillum, Rhodomicrobium* |
| | *bad*H | 2-hydroxycyclohexanecarboxyl-CoA dehydrogenase [EC:1.1.1.-] | *Cupriavidus, Magnetospirillum, Nocardioides* | *Alicyclobacillus, Aromatoleum, Bordetella, Rhodopseudomonas* |
| | *bad*I | 2-ketocyclohexanecarboxyl-CoA hydrolase [EC:3.1.2.-] | *nd* | *Cupriavidus, Leptothrix, Polaromonas, Xanthobacter* |
| | *hba*A | 4-hydroxybenzoate-CoA ligase [EC:6.2.1.27 6.2.1.25] | *Bradyrhizobium, Magnetospirillum, Rhodopseudomonas* | *Bradyrhizobium* |
| | *hba*B, *hcr*C | 4-hydroxybenzoyl-CoA reductase subunit gamma [EC:1.3.7.9] | *Magnetospirillum, Rhodopseudomonas* | *nd* |
| | *hba*C, *hcr*A | 4-hydroxybenzoyl-CoA reductase subunit alpha [EC:1.3.7.9] | *Rhodopseudomonas, Thauera* | *nd* |
| | *lig*AB | protocatechuate 4,5-dioxygenase [EC 1.13.11.8] | *Rhodopseudomonas, Acidimicrobium, Burkholderia, Nakamurella, Novosphingobium, Pseudoalteromonas, Xanthomonas* | *Brevundimonas, Azoarcus, Brevundimonas, Marinomonas, Saccharomonospora* |
| | *lig*C | 2-hydroxy-4-carboxymuconate semialdehyde hemiacetal dehydrogenase [EC 1.1.1.312] | *Rhodopseudomonas* | *Brevundimonas* |
| | *lig*I | 2-pyrone-4,6-dicarboxylate lactonase [EC:3.1.1.57] | *Agrobacterium, Albidiferax, Arthrobacter, Bradyrhizobium, Novosphingobium, Rhodopseudomonas* | *Agrobacterium, Arthrobacter, Bradyrhizobium, Delftia, Polaromonas, Rhodopseudomonas* |
| | *lig*J | 4-oxalomesaconate hydratase [EC 4.2.1.83] | *Arthrobacter, Asticcacaulis, Bradyrhizobium, Brevundimonas, Burkholderia, Cupriavidus, Burkholderia, Methylobacterium, Paracoccus, Pseudoalteromonas, Ralstonia, Rhizobium, Rhodopseudomonas, Sphingobium, Variovorax, Xanthomonas* | *Azoarcus,Brevundimonas, Burkholderia, Magnetospirillum,Methylobacterium, Novosphingobium, Paracoccus, Polaromonas, Rhizobium, RhodopseudomonasVerminephrobacter, Xanthomonas* |
| | *pob*A | p-hydroxybenzoate 3-monooxygenase [EC:1.14.13.2] | *Amycolatopsis, Arthrobacter, Bacillus, Burkholderia, Candidatus, Caulobacter, Chelativorans Delftia, Frankia, Herbaspirillum, Mesorhizobium, Novosphingobium, Rhodopseudomonas, Sphaerobacter, Streptosporangium* | *Agrobacterium, Amycolatopsis, Arthrobacter, Burkholderia, Candidatus, Magnetospirillum, Pseudomonas, Rhodopseudomonas, Saccharomonospora, Xanthomonas, Verminephrobacter* |
| | *pca*G | beta-Carboxy-cis,cis-muconate [EC:1.13.11.3] | *Bradyrhizobium, Candidatus, Cupriavidus, Geodermatophilus, Sphaerobacter, Sphingomonas* | *Candidatus* |
| | *pca*H | protocatechuate 3,4-dioxygenase, beta subunit [EC:1.13.11.3] | *Acidiphilium, Amycolatopsis, Arthrobacter, Azospirillum, Burkholderia, Caulobacter, Klebsiella, Pirellula, Planctomyces, Polaromonas, Pseudomonas, Rhodopirellula, Rubrobacter, Saccharopolyspora, Spirosoma, Streptosporangium, Xanthobacter* | *Agrobacterium, Azorhizobium, Azospirillum, Burkholderia, Candidatus, Cupriavidus, Geodermatophilus, Klebsiella, Mesorhizobium, Planctomyces, Polaromonas, Pseudomonas, Rhodococcus, Rhodopirellula, Saccharopolyspora, Spirosoma, Streptomyces, Streptosporangium, Xanthomonas* |
| | *pca*I | 3-oxoadipate CoA-transferase, alpha subunit [EC:2.8.3.6] | *Achromobacter, Arthrobacter, Kocuria, Serratia, Shewanella, Xanthomonas* | *Catenulispora, Mycobacterium* |
| | *pca*J | 3-oxoadipate CoA-transferase, beta subunit [EC:2.8.3.6] | *Arthrobacter, Cupriavidus, Escherichia, Herbaspirillum* | *Nocardioides, Rhodopseudomonas Ruegeria, Streptomyces* |
| | *pca*B | 3-carboxy-cis, cis-muconate cycloisomerase [EC:5.5.1.2] | *Cupriavidus, Deinococcus, Delftia, Frankia, Pseudomonas, Sinorhizobium, Xanthomonas* | *Albidiferax, Burkholderia, Chelativorans, Cupriavidus, Leptothrix, Meiothermus, Methylobacterium, Polaromonas, Ralstonia , Xanthomonas* |
| | *pca*C | 4-carboxymuconolactone decarboxylase [EC:4.1.1.44] | *Beutenbergia, Bradyrhizobium, Burkholderia, Candidatus, Methylobacterium, Mycobacterium, Rhizobium, Rhodomicrobium, Sinorhizobium, Yersinia* | *Amycolatopsis, Arthrobacter, Bradyrhizobium, Burkholderia, Frankia, Mycobacterium, Ralstonia, Rhodococcus, Rhodopseudomonas, Sinorhizobium* |
| | *pca*D | 3-oxoadipate enol-lactonase [EC:3.1.1.24] | *Burkholderia, Rhizobium, Xanthobacter* | *Chelativorans, Cupriavidus, Geodermatophilus, Methylobacterium, Mycobacterium, Rhizobium, Sinorhizobium, Thermomonospora* |
| | *pca*F | 3-oxoadipyl-CoA thiolase [EC:2.3.1.174] | *Bradyrhizobium, Pseudomonas, Rhodopseudomonas* | *Bradyrhizobium, Sphingomonas* |
| | *cat*A | catechol 1,2-dioxygenase [EC:1.13.11.1] | *Cupriavidus, Methylobacterium, Pseudomonas, Sphingobium, Streptomyces* | *Azospirillum, Bradyrhizobium, Burkholderia, Delftia,Mycobacterium, Pseudomonas, Rhizobium, Sphingomona, Xanthobacter* |
| | *cat*B | muconate cycloisomerase [EC:5.5.1.1] | *Rhodopirellula* | *Mycobacterium* |
| | *cat*E/ *dmp*B | catechol 2,3-dioxygenase [EC:1.13.11.2] | *Agrobacterium, Bradyrhizobium, Burkholderia, Meiothermus, Methylibium, Rhodococcus, Thauera* | *Arthrobacter, Bradyrhizobium, Brucella, Geobacillus, Meiothermus, Rubrobacter* |

2, 3-diol, 1, 2-dioxygenase (*bph*C) is another important enzyme that was noted in both the metagenome. The *bph*C genes were assigned to *Mycobacterium* in MGB-2, whereas *Alkalilimnicola*, *Mycobacterium*, and *Polynucleobacter* in MGB-3. Metabolic pathways of polychlorinated biphenyls degradation by *Pseudomonas*, *Polaromonas*, *Mycobacterium*, and *Polynucleobacter* have been studied in detail [12, 22, 35, 37, 44, 45].

Further, the annotated gene/enzyme and the bacterial genera involved in the benzoate degradation pathway were identified in MGB-2 and MGB-3 as listed in Table 4C. The presence of benzoate ligase in MGB-2 indicated anaerobic degradation of benzoate. Benzoate is the most common intermediate in the metabolism of aromatic compounds, and in the anaerobic condition, it is converted into benzoyl-CoA by benzoate ligase [46]. The presence of genes encoding anaerobic degradation *via* benzoyl CoA (*badA*, *badD*, *badE*, *badF*, *badH*, *badI*, *hbaA*, *hbaB*, *hcrC*) and aerobic *via* protocatechuate (*ligAB*, *ligC*, *ligI*, *ligJ*, *pobA*, *pcaB*, *pcaC*, *pcaDF*, *pcaGH* and *pcaIJ*) catechol (*catE/ dmpB*) indicates complete pathway for degradation of benzoate. Protocatechuate can be mineralized *via* ortho- cleavage by protocatechuate 3, 4-dioxygenase (PcaGH), and meta cleavage by protocatechuate 4, 5-dioxygenase (LigAB). Anaerobic degradation *via* benzoyl-CoA has also been documented in a variety of facultative anaerobes, including the denitrifying *Thauera* [47], *Magnetospirillum* strains [48], and the photoheterotroph *Rhodopseudomonas* [49]. These enzymes are contributed possibly by members of genera *Aromatoleum*, *Arthrobacter*, *Burkholderia*, *Bradyrhizobium*, *Cupriavidus*, *Magnetospirullum*, *Methylobacterium*, *Rhodomicrobium*, *Rhodopseudomonas*, and *Polaromonas* in both the metagenome communities suggesting a synergistic degradation of benzoate. Analysis of functional gene annotation using SEED subsystem by STAMP for the presence of peripheral degradation pathway in MGB-2 and MGB-3 revealed the abundance of, muconate cycloisomerase ring hydroxylating dioxygenase, naphthalene dioxygenase, acetaldehyde dehydrogenase, 2-hydroxy-6-oxo-6-phenyl hexa-2,4-dienoate hydrolase, and benzoate transport protein in MGB-2. On the other hand, genes encoding biphenyl 2,3 diol dioxygenase, benzaldehyde dehydrogenase, *nap/bph* dioxygenase, ortho-halobenzoate 1,2- dioxygenase, phenol dioxygenase, and 1,2-dihydroxycyclohexa-3,5-diene-1-carboxylate dehydrogenate were abundant in MGB-3 (S7 Fig). Similarly, a two-way comparison between MGB-2 and MGB-3 of functional gene annotation using SEED subsystem for aromatic degradation revealed significant level of abundance for benzoate ligase ($p = 2.63\ e^{-3}$) and 4-hydroxyphenylacetate 3-monooxygenase ($p = 8.93\ e^{-3}$) in MGB-2. The presence of 2-hydroxycyclohexanecarboxyl dehydrogenase with $p = 2.39\ e^{-3}$ indicated anaerobic degradation of benzoate in MGB-3 as well. However, aerobic degradation of biphenyl was also found in MGB-3 which is evident by the abundance of biphenyl- 2, 3-diol 1, 2-dioxygenase with $p = 0.035$ in MGB-3 (S8 Fig). Furthermore, a comparison between MGB-2 and MGB-3 at functional gene annotation using RefSeq for aromatic degradation revealed a significant level of abundance of *Acidothermus* ($p = 2.68\ e^{-3}$), *Nakamurella* ($p = 6.39\ e^{-3}$) in MGB-2 while *Mycobacterium* ($p = 3.30\ e^{-3}$), *Leptothrix* ($p = 7.89\ e^{-3}$) and *Polynuclearbacter* ($p = 0.034$) in MGB-3 (S9 Fig).

Our findings suggest that biphenyl (PCB) and pyrene/phenanthrene (PAH) biodegradation pathways could be linked together *via* a common intermediate protocatechuate pathway and undergoes complete degradation through the common protocatechuate branch of the β-ketoadipate pathway. Therefore, the reconstruction of complete biphenyl/PCB and PAH degradative pathways, based on the annotated genes, were done and shown in Fig 5. Cytoscape-based networking revealed microbial interaction in the xenobiotic biodegradation pathway. The key biodegraders in xenobiotic biodegradation pathways (aromatic halogenated, dioxin, PAH, PCB/biphenyl, catechol, protocatechuate, benzoyl-CoA) were found to be *Arthrobacter chlorophenolicus*, *Meiothermus ruber*, *Cupriavidus metallidurans*, *Burkholderia xenovorans*, *Rubrobacter xylanophilus*, *Bradyrhizobium* sp. BTAi1, *Bradyrhizobium japonicum*, *Sphingobium*

**Fig 5. Reconstruction of complete Biphenyl/P.C.B. and PAH degradation pathways based on annotated genes identified.** Blue, biphenyl degradation; red, benzoate degradation *via* catechol; pink, benzoate degradation via protocatechuate; orange, benzoate degradation *via* benzoyl CoA degradation; Black, PAH (Pyrene and Phenanthrene) degradation enter central pathway *via* protocatechuate intermediate.

*japonicum*, *Pseudomonas aeruginosa*, *Polaromonas* sp. JS666 and *Rhizobium leguminosarum* in MGB-2 (Fig 6A). In MGB-3 it was *Bradyrhizobium japonicum*, *Rhizobium leguminosarum*, *Mycobacterium* sp.KMS, *Polaromonas* sp. JS666, *Rhodopseudomonas palustris*, *Xanthobacter autotrophicus*, *Mycobacterium smegmatis*, *Burkholderia cenocepacia*, and *Burkholderia xeno-vorans* (Fig 6B). The greater abundance of these genes and genera in sample MGB-2 compared to sample MGB-3 suggests a higher degrading capacity in sample MGB-2. However, the key biodegraders were *Bradyrhizobium*, *Burkholderia*, *Mycobacterium*, and *Rhodopseudomonas* in both the metagenome.

## Conclusions

The present metagenomic study highlighted microbial function annotation, extensive degradation capabilities in terms of xenobiotic degradation pathways and correlated with the presence of PAH and PCB in the contaminated soil of steel plants. In addition, physicochemical profiling of the soil samples provided valuable information regarding the presence of organic (C/N/P), inorganic nutrient (Ca, K, Mg, Na, Mn), and metal (Fe, Mn, Cu, Zn) present that can be an essential parameter for designing biodegradation strategies. Higher proportions of Proteobacteria and Actinobacteria indicated the two samples possess good biodegradation potential. Moreover, the coordination among different biodegraders and the presence of functional genes involved in biodegradation pathways and energy metabolism has provided an in-depth understanding of their survival under stress conditions of persistent organic pollutants. Moreover, these potential biodegraders such as *Bradyrhizobium*, *Burkholderia*, *Mycobacterium*, *Rhodopseudomonas*, and *Pseudomonas* identified in the present study can be further selected and further could be exploited for enhancing bioremediation. Therefore, it can be concluded that

A.

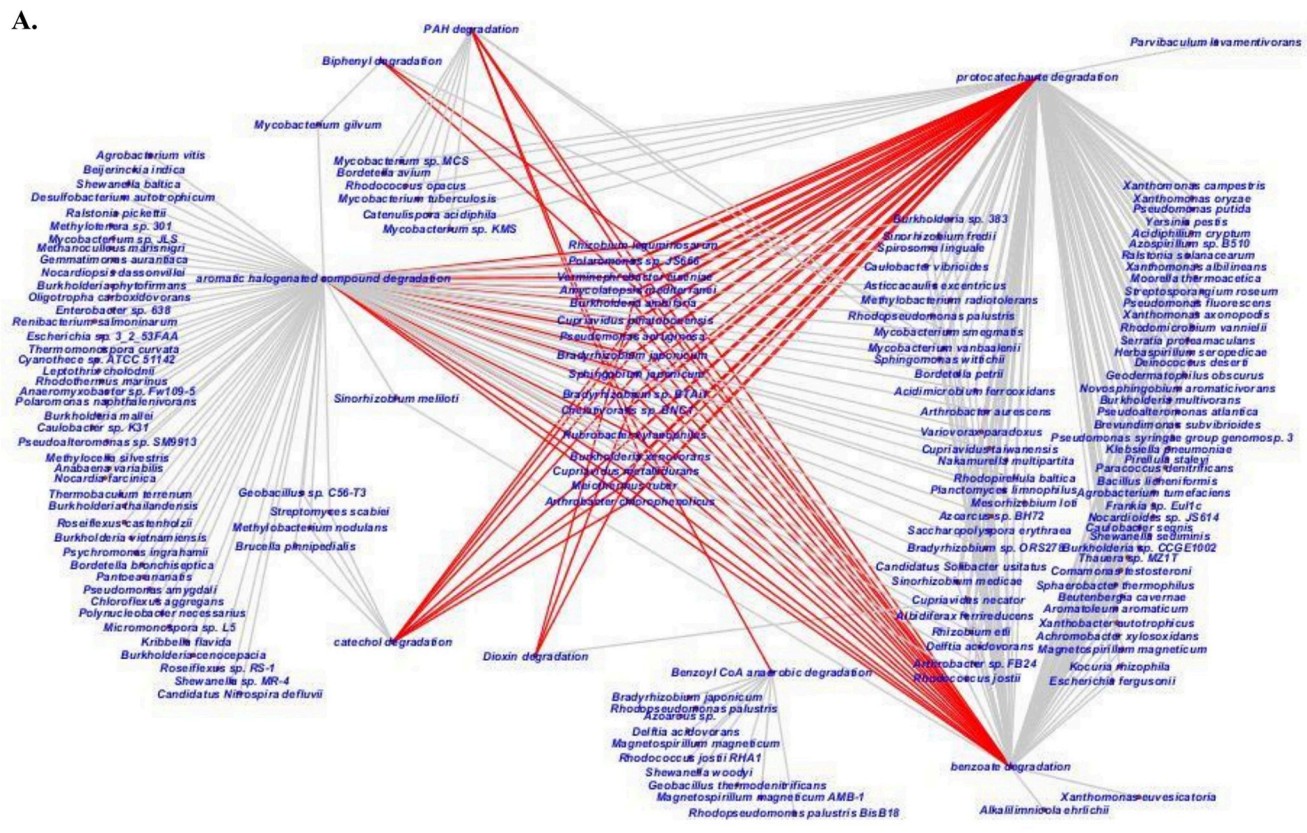

B.

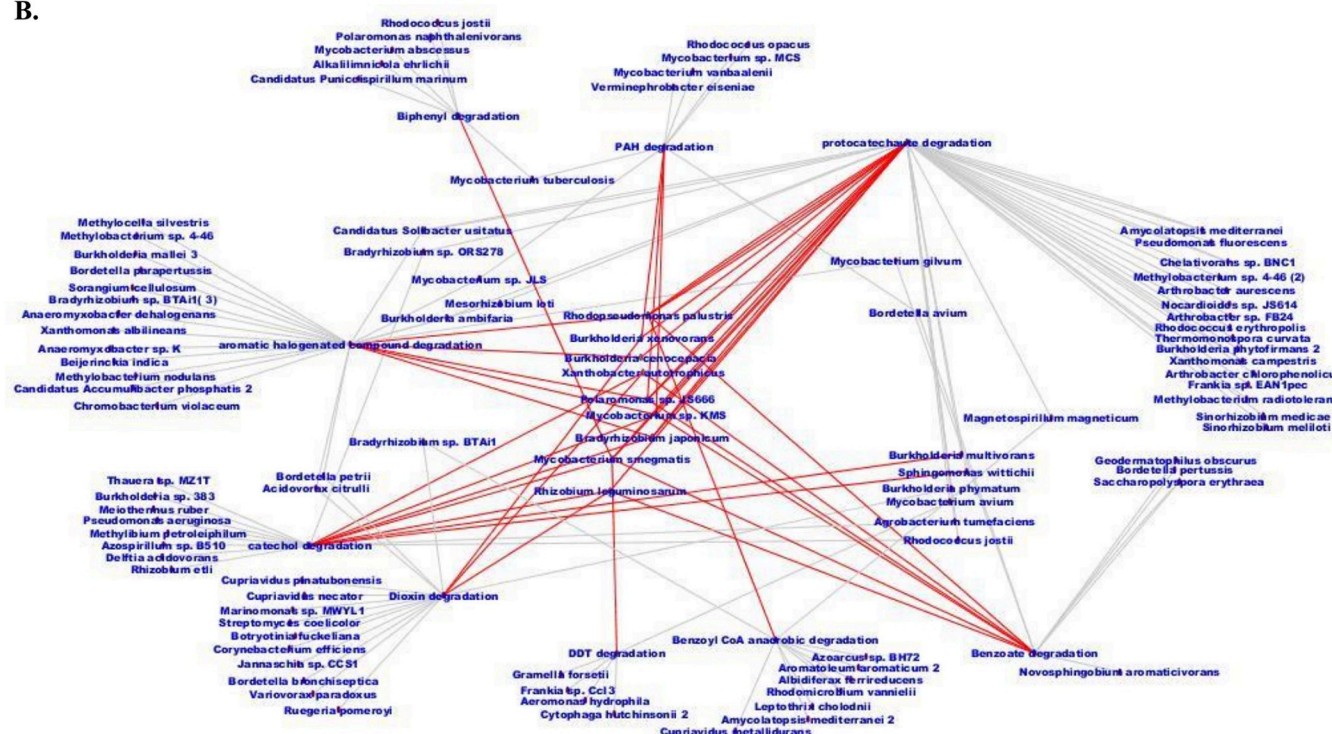

**Fig 6.** Cytoscape-based networking depicted interrelationship of key biodegraders in PAH, biphenyl, dioxin, halogenated, catechol and protocatechuate pathways Key biodegraders presented in the middle and the interrelated pathways are given in red lines (A) MGB-2 (B) MGB-3.

investigating microbial community and exploring their potential for biodegradation is a critical factor in maximizing the efficacy of the bioremediation process.

## Supporting information

**S1 Table. Barcodes used for sequencing.**
(JPG)

**S2 Table. The total number of unassembled and assembled reads used for downstream analyses.**
(JPG)

**S1 Fig. PCB identified in MGB-2 sample.**
(JPG)

**S2 Fig. PCB identified in MGB-3 sample.**
(JPG)

**S3 Fig. Rarefaction curve represented species richness indicating function of number of reads (y-axis) per MGB-2 and MGB-3 (x- axis).**
(JPG)

**S4 Fig. The scatter plot indicated a linear relationship between MGB-2 and MGB-3.**
(JPG)

**S5 Fig.** Comparison between MGB-2 and MGB-3 of functional gene annotation using STAMP using SEED subsystem a.) Tol and Ton b.) two-component system c.) membrane transport d.) Phosphate transporter.
(JPG)

**S6 Fig. Annotated pathways for xenobiotic degradation and metabolism accounted for metagenome MGB-2 and MGB-3.**
(JPG)

**S7 Fig. Comparative analysis of metagenome MGB-2 and MGB-3 based on Peripheral aromatic degradation metabolism at SEED subsystem level by using STAMP.**
(JPG)

**S8 Fig. Comparative analysis of metagenome MGB-2 and MGB-3 based on biphenyl degradation metabolism SEED level by using STAMP.**
(JPG)

**S9 Fig. Comparative analysis of metagenome MGB-2 and MGB-3 based on aromatic degradation metabolism at ReFSeq genus level by using STAMP.**
(JPG)

**S1 File.**
(DOCX)

## Acknowledgments

MS acknowledges financial assistance provided by Department of Science and Technology, New Delhi, India under Women Scientist Scheme B [SR/WOS-B/570/2016], and BITS Pilani. Our special thanks to Central Instrumentation Facility, Birla Institute of Technology and Sciences (BITS), Pilani campus for providing GC-MS/MS facility.

## Author Contributions

**Conceptualization:** Monika Sandhu, Atish T. Paul, Prabhat N. Jha.

**Data curation:** Atish T. Paul.

**Formal analysis:** Atish T. Paul.

**Investigation:** Monika Sandhu.

**Methodology:** Monika Sandhu.

**Software:** Monika Sandhu.

**Supervision:** Prabhat N. Jha.

**Visualization:** Atish T. Paul.

**Writing – original draft:** Monika Sandhu.

**Writing – review & editing:** Monika Sandhu, Atish T. Paul, Prabhat N. Jha.

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
