## [Decision Letter · Decision Letter 0]

3 Mar 2022

PONE-D-21-32138Metagenomic analysis for taxonomic and functional potential of Polyaromatic hydrocarbons (PAHs) and Polychlorinated biphenyl (PCB) degrading bacterial communities in steel industrial soilPLOS ONE

Dear Dr. Jha,

Thank you for submitting your manuscript to PLOS ONE. After careful consideration, we feel that it has merit but does not fully meet PLOS ONE’s publication criteria as it currently stands. Therefore, we invite you to submit a revised version of the manuscript that addresses the points raised during the review process.

We look forward to receiving your revised manuscript.

Kind regards,

Rachel S. Poretsky, PhD

Academic Editor

PLOS ONE

Journal Requirements:

2. We note that Figure (1) in your submission contain copyrighted images. All PLOS content is published under the Creative Commons Attribution License (CC BY 4.0), which means that the manuscript, images, and Supporting Information files will be freely available online, and any third party is permitted to access, download, copy, distribute, and use these materials in any way, even commercially, with proper attribution. For more information, see our copyright guidelines: http://journals.plos.org/plosone/s/licenses-and-copyright.

1. You may seek permission from the original copyright holder of Figure (1) to publish the content specifically under the CC BY 4.0 license. 

Reviewers' comments:

Reviewer's Responses to Questions

**Comments to the Author**

1. Is the manuscript technically sound, and do the data support the conclusions?

Reviewer #1: Partly

Reviewer #2: Yes

2. Has the statistical analysis been performed appropriately and rigorously? 

Reviewer #1: Yes

Reviewer #2: Yes

3. Have the authors made all data underlying the findings in their manuscript fully available?

Reviewer #1: No

Reviewer #2: Yes

4. Is the manuscript presented in an intelligible fashion and written in standard English?

Reviewer #1: Yes

Reviewer #2: Yes

5. Review Comments to the Author

Reviewer #1: Why did the authors choose to study the PAH and PCB degrading genes in soil collected from steel industry? What is the scientific relevance of a steel plant? Why not soil/sludge from petrochemical refining units?

What is the scientific evidence that the communities identified have PAH/PCB degradation potential? There is none mentioned/presented in this study. Presence of a PAH degrading gene doesn't confirm its expression.

Presence of a pollutant in soil doesn't confirm its degradation by the microbial communities present there? The growth dynamics. enzyme activity and gene expression in addition to PAH/PCB degradation can only confirm it. The idea lacks scientific evidence.

Reviewer #2: This manuscript presents an exhaustive community identification process that gives a unique outlook on PAH and PCB degradation in steel industry-contaminated site. My recommendation for this work is acceptance only after the minor issues highlighted in the reviewer document are addressed.

6. PLOS authors have the option to publish the peer review history of their article (what does this mean?). If published, this will include your full peer review and any attached files.

Reviewer #1: No

Reviewer #2: No

---

## [Author Response · Author response to Decision Letter 0]

17 Mar 2022

All the corrections have been done as per the Editor and reviewer's comment and is uploaded as response to reviewer's document

---

## [Editor Report · Decision Letter 1]

29 Mar 2022

Metagenomic analysis for taxonomic and functional potential of Polyaromatic hydrocarbons (PAHs) and Polychlorinated biphenyl (PCB) degrading bacterial communities in steel industrial soil

PONE-D-21-32138R1

Dear Dr. Jha,

We’re pleased to inform you that your manuscript has been judged scientifically suitable for publication and will be formally accepted for publication once it meets all outstanding technical requirements.

Kind regards,

Rachel S. Poretsky, PhD

Academic Editor

PLOS ONE

Additional Editor Comments (optional):

Journal Requirements:

1. In your Methods section, please provide additional information regarding the permits you obtained for the work. Please ensure you have included the full name of the authority that approved the field site access and, if no permits were required, a brief statement explaining why.
---

## [Editor Report · Acceptance letter]

6 Apr 2022

PONE-D-21-32138R1 

Metagenomic analysis for taxonomic and functional potential of Polyaromatic hydrocarbons (PAHs) and Polychlorinated biphenyl (PCB) degrading bacterial communities in steel industrial soil 

Dear Dr. Jha:

I'm pleased to inform you that your manuscript has been deemed suitable for publication in PLOS ONE. Congratulations! Your manuscript is now with our production department. 

Kind regards, 

on behalf of

Dr. Rachel S. Poretsky 

Academic Editor

PLOS ONE